# LEARNING WHEN AND WHAT TO ASK: A HIERARCHICAL REINFORCEMENT LEARNING FRAMEWORK

## ABSTRACT

Reliable AI agents should be mindful of the limits of their knowledge and consult humans when sensing that they do not have sufficient knowledge to make sound decisions. We formulate a hierarchical reinforcement learning framework for learning to decide *when* to request additional information from humans and *what* type of information would be helpful to request. Our framework extends partially-observed Markov decision processes (POMDPs) by allowing an agent to interact with an assistant to leverage their knowledge in accomplishing tasks. Results on a simulated human-assisted navigation problem demonstrate the effectiveness of our framework: aided with an interaction policy learned by our method, a navigation policy achieves up to a $7\times$ improvement in task success rate compared to performing tasks only by itself. We find that the ability to request subgoals enables the agent to generalize effectively to tasks in unseen environments. We analyze benefits and challenges of learning with a hierarchical policy structure and suggest directions for future work.

## 1 INTRODUCTION

Human-agent communication at deployment time has been under-explored in machine learning, where the traditional focus has been on building agents that can accomplish tasks on their own (full autonomy). Nevertheless, enabling an agent to exchange information with humans during its operation can potentially enhance its helpfulness and trustworthiness. The ability to request and interpret human advice would help the agent accomplish tasks beyond its built-in knowledge, while the ability to accurately convey when and why it is about to fail would make the agent safer to use.

In his classical work, Grice (1975) outlines the desired characteristics of cooperative communication, commonly known as the *Gricean maxims of cooperation*. Among these characteristics are informativeness (the maxim of quantity) and faithfulness (the maxim of quality). Human-agent communication in current work has fallen short in these two aspects. Traditional frameworks like imitation learning and reinforcement learning employ limited communication protocols where the agent and the human can only exchange simple intentions (requesting low-level actions or rewards Torrey & Taylor (2013); Knox & Stone (2009)). More powerful frameworks like (Nguyen & Daumé III, 2019; Nguyen et al., 2019; Kim et al., 2019) allow the agent to process high-level instructions from humans, but the agent still only requests generic help. Recent work in natural language processing endows the agent with the ability to generate rich natural language utterances (Camburu et al., 2018; Rao & Daumé III, 2018; De Vries et al., 2017; Das et al., 2017), but the communication is not faithful in the sense that the agent only mirrors human-generated utterances without grounding its communication in self-perception of its (in)capabilities and (un)certainties. Essentially it learns to convey what a human may be concerned about, not what *it* is concerned about.

This paper presents a hierarchical reinforcement learning framework named HARI (**H**uman-**A**ssisted **R**einforced **I**nteraction), which supports richer and more faithful human-agent communication. Our framework allows the agent to learn to convey intrinsic needs for specific information and to incorporate diverse types of information from humans to make better decisions. Specifically, the agent in HARI is equipped with three information-seeking intentions: in every step, it can choose to request more information about (i) its current state, (ii) the goal state, or (iii) a subgoal state which, if reached, helps it make progress on the current task. Upon receiving a request, the human can transfer new information to the agent by giving new descriptions of the requested state. These de-

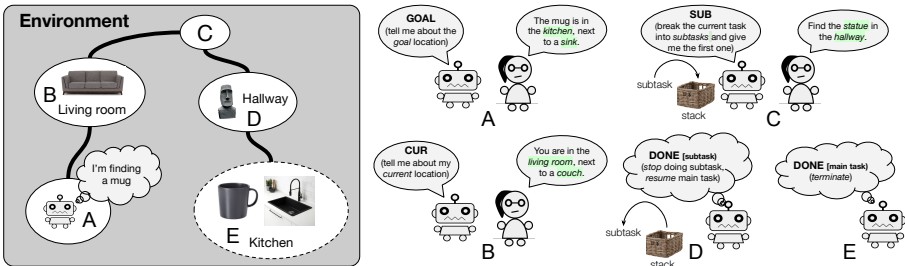

Figure 1: An illustration of the HARI framework in a human-assisted navigation task. An agent can only observe part of an environment and is asked to find a mug in the kitchen. An assistant communicates with the agent and can provide it with information about the environment and the task. Initially (A) it may request more information about the goal, but may not know enough about where it currently is. For example, at location B, due to limited perception, it does not recognize that it is in a living room and stands next to a couch. It can obtain such information from the assistant. If the current task becomes too difficult (like at location C), the agent can require the assistant to provide a simpler subtask which, if accomplished, helps it make progress on the main task. The agent maintains a stack of tasks and always executes the task at the top. When the agent receives a (sub)task, it pushes the (sub)task to the top of the stack. When it wants to stop executing a (sub)task, it pops the (sub)task from the stack (e.g., at location D). At location E, the agent empties the stack and terminates its execution.

scriptions will be incorporated as new inputs to the agent's decision-making policy. The human thus can transfer any form of information that can be interpreted by the policy (e.g., asking the agent to execute skills that it has learned, giving side information that connects the agent to a situation it is more familiar with). Because the agent's policy can implement a variety of model architectures and learning algorithms, our framework opens up many possibilities for human-agent communication.

To enable faithful communication, we teach the agent to understand its intrinsic needs by interacting with the human and the environment (rather than imitating human behaviors). By requesting different types of information and observing how much each type of information enhances its decisions, the agent gradually learns to determine which information is most useful to obtain in a given situation. With this capability, at deployment time, it can choose when and what information to ask from the human to improve its task performance. To demonstrate the effectiveness of HARI, we simulate a human-assisted navigation problem where an agent has access to only sparse information about its current state and the goal, and can request additional information about these states. On tasks that take place in previously unseen environments, the ability to ask for help improves the agent's success rate by $7\times$ higher compared to performing tasks only by itself. This human-assisted agent even outperforms an agent that always has access to dense information in unseen environments, thanks to the ability to request subgoals. We show that performance of the agent can be further improved by recursively asking for subgoals of subgoals. We discuss limitations of the policy's model and feature representation, which suggest room for future improvements.

## 2 MOTIVATION: LIMITATIONS OF THE STANDARD POMDP FRAMEWORK

We consider an environment defined by a partially-observed Markov decision process (POMDP) $E = (\mathcal{S}, \mathcal{A}, T, c, \mathcal{D}, \rho)$ with state space $\mathcal{S}$, action space $\mathcal{A}$, state-transition function $T : \mathcal{S} \times \mathcal{A} \to \Delta(\mathcal{S})$, cost function $c : \mathcal{S} \times \mathcal{A} \to \mathbb{R}$, description space $\mathcal{D}$, and description function $\rho : \mathcal{S} \to \Delta(\mathcal{D})$.[1] Here, $\Delta(\mathcal{Y})$ denotes the set of all probability distributions over a set $\mathcal{Y}$. We refer to this environment as the *operation environment* because it is where the agent operates to accomplish tasks.

Each *task* in the environment is defined as a tuple $(s_1, g_1, d_1^g)$ where $s_1$ is the start state, $g_1$ is the goal state, and $d_1^g$ is a limited description of $g_1$. Initially, a task $(s_1, g_1, d_1^g)$ is sampled from a task

---

[1] We use the term "description" in lieu of "observation" in the POMDP formulation to emphasize two properties of the information the agent has access to for making decisions: (i) the information can be in various modalities and (ii) the information can be obtained via not only perception, but also communication.

distribution $\mathfrak{T}$. An agent starts in $s_1$ and is only given the goal description $d_1^g$. It has to reach the goal state $g_1$ within $H$ time steps. Let $g_t$ and $d_t^g$ be the goal state and goal description being executed at time step $t$, respectively. In a standard POMDP, $g_t = g_1$ and $d_t^g = d_1^g$ for $1 \leq t \leq H$. But later, we will enable the agent to set new goals via communication with humans.

At any time step $t$, the agent does not know its true state $s_t$ but only receives a *description* $d_t^s \sim \rho(s_t)$ of the state. Generally, the description can include any information coming from any knowledge source (e.g., an RGB image and/or a verbal description describing the current view). Given $d_t^s$ and $d_t^g$, the agent then makes a decision $a_t \in \mathcal{A}$, transitions to the next state $s_{t+1} \sim T(s_t, a_t)$, and receives a cost $c_t \triangleq c(s_t, a_t)$. A special action $a_{\text{done}} \in \mathcal{A}$ is taken when the agent decides to terminate its execution. The goal of the agent is to reach $g_1$ with minimum cumulative cost $C(\tau) = \sum_{t=1}^H c_t$, where $\tau = (s_1, d_1^s, a_1, s_2, d_2^s, a_2, \ldots, s_H, d_H^s)$ is an execution of the task.

As the agent does not have access to its true state, it can only make decisions based on the (observable) partial execution $\tau_{1:t} = (d_1^s, a_1, \ldots, d_t^s)$. Kaelbling et al. (1998) introduce the notion of a *belief state* $b \in \Delta(\mathcal{S})$, which sufficiently summarizes a partial execution as a distribution over the state space $\mathcal{S}$. In practice, when $\mathcal{S}$ is continuous or high-dimensional, representing and updating a full belief state (whose dimension is $|\mathcal{S}|$) is intractable. We follow Hausknecht & Stone (2015), using recurrent neural networks to learn compact representation of partial executions. We denote by $b_t^s$ a representation of the partial execution $\tau_{1:t}$ and by $\mathcal{B}$ the set of all possible representations.

The agent maintains an *operation policy* $\hat{\pi} : \mathcal{B} \times \mathcal{D} \to \Delta(\mathcal{A})$ that maps a belief state $b^s$ and a goal description $d^g$ to a distribution over $\mathcal{A}$. The learning objective for solving a standard POMDP is to estimate an operation policy that minimizes the expected cumulative cost of performing tasks:

$$\min_\pi \mathbb{E}_{(s_1, g_1, d_1^g) \sim \mathfrak{T}, \tau \sim P_\pi(\cdot | s_1, d_1^g)} [C(\tau)] \tag{1}$$

where $P_\pi(\cdot \mid s_1, d_1^g)$ is the distribution over executions generated by a policy $\pi$ given start state $s_1$ and goal description $d_1^g$. In a standard POMDP, an agent performs tasks by executing its own operation policy without asking for any external assistance. Moreover, the description function $\rho$ and the goal description $d_1^g$ are assumed to be fixed during a task execution. As seen from Equation 1, given a fixed environment and task distribution, the expected performance of the agent is solely determined by the operation policy $\hat{\pi}$. Thus, the standard POMDP framework does not provide any mechanism for improving the agent's performance other than enhancing the operation policy.

## 3 LEVERAGING HUMAN KNOWLEDGE TO BETTER ACCOMPLISH TASKS

We introduce an *assistant* into the operation environment, who can provide information about the environment's states. We assume the agent possesses a pre-learned operation policy $\hat{\pi}$. This policy serves as the *common ground* between the agent and the assistant, which is a prerequisite for communication between them to occur. For example, this policy represents a set of basic tasks that agent has mastered and the assistant can ask the agent to perform. In general, the more knowledge encoded in this policy, the more effectively the agent can communicate with and leverage help from the assistant. Our goal is to learn an *interaction policy* $\psi_\theta$ (parametrized by $\theta$) that controls how the agent communicates with the assistant to gather additional information. The operation policy $\hat{\pi}$ will be invoked by the interaction policy if the latter decides that the agent does not need new information and wants to take an operating action.

The assistant aids the agent by giving new (current or goal) state descriptions, connecting the agent to situations on which it can make better decisions. Consider an object-finding navigation problem, where a robot has been trained to reliably navigate to the kitchen from the living room of a house. Suppose the robot is then asked to "*find a mug*", an object that it has never heard of. The assistant can help the robot accomplish this task by giving a more informative goal description "*find a mug in the kitchen*", relating the current task to the kitchen-finding task that the robot has been familiar with. The robot may also have problems with localization: it knows how to get the kitchen from the living room but it may not realize that it is currently the living room. In this case, giving a current-state description that specifies this information provides the robot with a useful hint on what actions to take next.

Our framework allows the assistant to convey any form of information that the agent can incorporate into its input. As discussed in §2, the notion of "state description" in our framework is general,

capturing various types of information, including but not limited to visual perception and verbal description. Communication between the agent and the assistant can be flexibly enriched by designing the agent's operation policy to be able to consume the forms of information of interest (e.g., a policy that takes natural language as input).

**Communication with the Assistant.** The assistant is present all the time and knows the agent's current state $s_t$ and the goal state $g_t$. It is specified by two functions: a description function $\rho_A : \mathcal{S} \times \mathcal{D} \to \Delta(\mathcal{D})$ and a subgoal function $\omega_A : \mathcal{S} \times \mathcal{S} \to \Delta(\mathcal{S})$. $\rho_A(d' \mid s, d)$ specifies the probability of giving $d'$ as the new description of state $s$ given a current description $d$. $\omega_A(g' \mid s, g)$ indicates the probability of proposing $g'$ as a subgoal given a current state $s$ and a goal state $g$.

At time step $t$, the assistant accepts three types of request from the agent:

(a) CUR: requests a new description of $s_t$ and receives $d_{t+1}^s \sim \rho_A\left(\cdot \mid s_t, d_t^s\right)$;
(b) GOAL: requests a new description of $g_t$ and receives $d_{t+1}^g \sim \rho_A\left(\cdot \mid g_t, d_t^g\right)$;
(c) SUB: requests a description of a subgoal $g_{t+1}$ and receives $d_{t+1}^g \sim \rho_A\left(\cdot \mid g_{t+1}, \emptyset\right)$ where $g_{t+1} \sim \omega_A\left(\cdot \mid s_t, g_t\right)$ and $\emptyset$ is an empty description.

**Interaction Policy.** The action space of the interaction policy $\psi_\theta$ consists of five actions: $\{$CUR, GOAL, SUB, DO, DONE$\}$. The first three actions correspond to making the three types of request that the assistants accepts. The remaining two actions are used to traverse in the operation environment:

(d) DO: executes the action $a_t^{\text{do}} \triangleq \arg\max_{a \in \mathcal{A}} \hat{\pi}\left(a \mid b_t^s, d_t^g\right)$. The agent transitions to a new operation state $s_{t+1} \sim T(s_t, a_t^{\text{do}})$;
(e) DONE: determines that the current goal $g_t$ has been reached.[2] If $g_t$ is a main goal ($g_t = g_1$), the task episode ends. If $g_t$ is a subgoal ($g_t \neq g_1$), the agent may choose a new goal to follow. Our problem formulation leaves it open on what goal should be selected next.

By selecting among these actions, the interaction policy essentially decides *when* to ask the assistant for additional information, and *what* types of information to ask for. Our formulation does not specify the input space of the interaction policy, as this space depends on how the agent implements its goal memory (i.e. how it stores and retrieves the subgoals). In the next section, we introduce an instantiation where the agent uses a stack data structure to manage (sub)goals.

## 4 HIERARCHICAL REINFORCEMENT LEARNING FRAMEWORK

In this section, we describe the HARI framework. We first formulate the POMDP environment that the interaction policy acts in, referred to as the *interaction environment* (§ 4.1). Our construction employs a *goal stack* to manage multiple levels of (sub)goals (§4.2). A goal stack stores all the tasks the agent has been assigned but has not yet decided to terminate (by choosing the DONE action). It is updated in every step depending on the taken action. We design a cost function (§ 4.4) that specifies a trade-off between the cost of taking actions (acting cost) and the cost of not completing a task (task error).

### 4.1 INTERACTION ENVIRONMENT

Given an operation environment $E = (\mathcal{S}, \mathcal{A}, T, c, \mathcal{D}, \rho)$, the interaction environment constructed on top of $E$ is a POMDP $\bar{E} = (\bar{\mathcal{S}}, \bar{\mathcal{A}}, \bar{\mathcal{T}}, \bar{c}, \mathcal{D}, \bar{\rho})$ with:

- State space $\bar{\mathcal{S}} = \mathcal{S} \times \mathcal{D} \times \mathcal{G}_L$ where $\mathcal{G}_L$ is the set of all goal stacks containing at most $L$ elements ($L$ is a hyperparameter). Each state $\bar{s} = (s, d^s, G) \in \bar{\mathcal{S}}$ is a tuple of an operation state $s \in \mathcal{S}$, its description $d^s \in \mathcal{D}$, and a goal stack $G \in \mathcal{G}_L$. Each element in the goal stack $G$ is a tuple $(g, d^g)$ of a goal state $g \in \mathcal{S}$ and its description $d^g \in \mathcal{D}$;
- Action space $\bar{\mathcal{A}} = \{$CUR, GOAL, SUB, DO, DONE$\}$;
- State-transition function $\bar{T} = T_s \cdot T_G$ where $T_s : \mathcal{S} \times \mathcal{D} \times \bar{\mathcal{A}} \to \Delta(\mathcal{S} \times \mathcal{D})$ and $T_G : \mathcal{G}_L \times \bar{\mathcal{A}} \to \Delta(\mathcal{G}_L)$;
- Cost function $\bar{c} : (\mathcal{S} \times \mathcal{G}_L) \times \bar{\mathcal{A}} \to \mathbb{R}$ (defined in §4.4 to trade off acting cost and task error);

---

[2]Note that the agent may falsely decide that a goal has been reached.

- Description space $\bar{\mathcal{D}} = \mathcal{D} \times \mathcal{G}_L^d$ where $\mathcal{G}_L^d$ is the set of all goal-description stacks of size $L$. At any time, the agent cannot access the environment's goal stack $G$, which contains true goal states. Instead, it can only observe the descriptions in $G$. We call this partial stack a *goal-description stack*, denoted by $G^d$;
- Description function $\bar{\rho} : \bar{\mathcal{S}} \to \bar{\mathcal{D}}$, where $\bar{\rho}(\bar{s}) = \bar{\rho}(s, d^s, G) = (d^s, G^d)$. Unlike in the standard POMDP formulation, this description function is deterministic.

A belief state $\bar{b}_t$ summarizes a partial execution $(\bar{s}_1, \bar{a}_1, \cdots, \bar{s}_t)$. We formally define the interaction policy as $\psi_\theta : \bar{\mathcal{B}} \to \Delta(\mathcal{A})$, where $\bar{\mathcal{B}}$ is the set of all interaction belief states.

## 4.2 GOAL STACK

A goal stack is an ordered set of tasks that the agent has not declared completion (by calling DONE). The initial stack $G_1 = \{(g_1, d_1^g)\}$ contains the main goal $g_1$, and its description $d_1^g$. Let $G_t$ be the goal stack at time step $t$. The agent executes the goal $g_t$ at the top of this stack. Only the GOAL, SUB, DONE actions alter the stack. The GOAL action replaces the top goal description with $d_{t+1}^g$, the new description given by the assistant. The SUB action pushes a new subtask $(g_{t+1}, d_{t+1}^g)$ to the stack. The DONE action pops the top (sub)task from the stack.

$$G_t.\texttt{update}(a) = \begin{cases} G_t.\texttt{pop()}.\texttt{push}(g_t, d_{t+1}^g) & \text{if } a = \text{GOAL}, \\ G_t.\texttt{push}(g_{t+1}, d_{t+1}^g) & \text{if } a = \text{SUB}, \\ G_t.\texttt{pop()} & \text{if } a = \text{DONE}, \\ G_t & \text{otherwise} \end{cases} \tag{2}$$

The SUB action is not available to the agent when the current stack contains $L$ elements, guaranteeing that goal stack always has at most $L$ elements. The goal-stack transition function $T_G$ is defined as $T_G(G_{t+1} \mid G_t, \bar{a}_t) = \mathbb{1}\{G_{t+1} = G_t.\texttt{update}(\bar{a}_t)\}$ where $\mathbb{1}\{.\}$ is an indicator function.

## 4.3 TRANSITION OF THE CURRENT OPERATION STATE AND ITS DESCRIPTION

To complete the definition of the state-transition function $\bar{T}$, we define the transition function $T_s$. This function is factored into two terms by the chain rule:

$$T_s(s_{t+1}, d_{t+1}^s \mid s_t, d_t^s, \bar{a}_t) = P(s_{t+1} \mid s_t, \bar{a}_t) \cdot P(d_{t+1}^s \mid s_{t+1}, d_t^s, \bar{a}_t) \tag{3}$$

Only taking the DO action may change the current operation state

$$P(s_{t+1} \mid s_t, \bar{a}_t) = \begin{cases} T\left(s_{t+1} \mid s_t, a_t^{\text{do}}\right) & \text{if } \bar{a}_t = \text{DO}, \\ \mathbb{1}\{s_{t+1} = s_t\} & \text{otherwise} \end{cases} \tag{4}$$

The description $d_t^s$ may vary when the agent moves to a new operation state (by taking the DO action) or requests a new description of $s_t$ (by taking the CUR action)

$$P(d_{t+1}^s \mid d_t^s, s_{t+1}, \bar{a}_t) = \begin{cases} \rho(d_{t+1}^s \mid s_{t+1}) & \text{if } \bar{a}_t = \text{DO}, \\ \rho_A(d_{t+1}^s \mid s_{t+1}, d_t^s) & \text{if } \bar{a}_t = \text{CUR}, \\ \mathbb{1}\{d_{t+1}^s = d_t^s\} & \text{otherwise} \end{cases} \tag{5}$$

## 4.4 COST FUNCTION

The interaction policy needs to balance between two types of cost: the cost of taking actions (acting cost) and the cost of not completing a task (task error). The acting cost also subsumes the cost of communicating with the assistant because, in reality, such interactions consume time, human effort, and possibly trust. Assuming that the assistant is helpful, acting cost and task error usually conflict with each other; for example, the agent may lower its task error if it is willing to suffer a larger acting cost by increasing the number of requests to the assistant.

We employ a simplified model where all types of cost are non-negative real numbers of the same unit. Making a request of type $a$ is assigned a constant cost $\gamma_a$. The cost of taking the DO action is $c(s_t, a_t^{\text{do}})$, the cost of executing the $a_t^{\text{do}}$ action in the operation environment. Calling DONE to terminate execution of the main goal $g_1$ incurs a task error $c(s_t, a_{\text{done}})$. We exclude the task errors

of executing subgoals because the interaction policy is only evaluated on reaching the main goal. The magnitudes of the costs naturally specify a trade-off between acting cost and task error. For example, setting the task errors much larger than the other costs indicates that completing tasks is prioritized over taking few actions.

## 5   LEARNING WHEN AND WHAT TO ASK IN HUMAN-ASSISTED NAVIGATION

**Problem.**   We apply HARI to modeling a human-assisted navigation (HAN) problem. In HAN, a human requests an agent to find an object in an indoor environment. Each task request asks the agent to go to a room of type $r$ and find an object of type $o$ (e.g., find a mug in a kitchen). The agent is equipped with a camera and shares its camera view with the human. We assume that the human is sufficiently familiar with the environment that they can recognize the agent's location by looking at its current view. Before issuing a task request, the human imagines a goal location (but do not reveal it to the agent). We are primarily interested in evaluating success in *goal-finding*, i.e. whether the agent can arrive at the human's intended goal location. Even though there could be multiple locations that match a request, the agent only succeeds if it arrives exactly at the chosen goal location. We also determine success in *request-fulfilling*, where the agent successfully fulfills a request if it navigates to any node that is within two meters of an object that matches the request.

While an agent is performing a task, it may request the human to provide additional information via telecommunication (e.g., a chat app). Specifically, it can ask for a description of its current location (CUR), the goal location (GOAL), or a subgoal location that is on the path from its current location to the goal location (SUB). Detail about how the subgoals are determined is in the Appendix.

**Operation Environment.**   We construct the operation environments using the environment graphs provided by the Matterport3D simulator (Anderson et al., 2018). Each environment graph is generated from a 3D model of a house where each node is a location in the house and each edge connects two nearby unobstructed locations. Each operation state $s$ corresponds to a node in the graph. At any time, the agent's operation action space $\mathcal{A}$ consists of traversing to any of the nodes that are adjacent to its current node.

We employ a discrete bag-of-features representation for state descriptions.[3] A bag of features represents the information that the agent extracts from the raw input that the agent perceives (e.g., an image, a language sentence). Working with this intermediate input allows us to easily vary the type and amount of information given to the agent. Specifically, we simulate two settings of descriptions: *dense* and *sparse*. At evaluation time, the agent perceives sparse descriptions and request the assistant for dense descriptions. A dense description of a current location contains the room name at the location, and the features of $M$ objects restricted to be within $\delta$ meters of the location. The features of each object consists of (i) its name, (ii) horizontal and vertical angles (relative to the current viewpoint), and (iii) distance (in meters) from the object to the current location. A dense description of a goal follows the same representation scheme. In the sparse setting, the current-location description does not include the room name. Moreover, we remove the features of objects that are not in the top 100 most frequent objects, emulating an imperfect object detector module. The sparse goal description (the task request) has only features of the target object and the room name where the object is located at. Especially, if a subgoal location is adjacent or coincides with the agent's current location, instead of describing room and object features, the human specifies the ground-truth action to go to the subgoal (an action is specified by its horizontal and vertical angles, and travel distance).[4]

**Experimental Procedure.**   We conduct our experiments in three phases. In the *pre-training* phase, we learn an operation policy $\hat{\pi}$ with dense descriptions of the current location and the goal. In the *training* phase, the agent perceives a sparse description of its current location and is given a sparse initial goal description. We use advantage actor-critic (Mnih et al., 2016) to learn an interaction policy $\psi_\theta$ that controls how the agent communicates with the human and navigates in an environment.

---

[3]While our representation of state descriptions simplifies the object/room detection problem for the agent, it does not necessarily make the navigation problem easier than with image input, as images may contain information that is not captured by our representation (e.g., object shapes and colors, visualization of paths).

[4]Here, we emulate a practical scenario where if a destination is visible in the current view, to save effort, a human would concisely tell an agent what to do rather than giving a verbose description of the destination.

Table 1: Main results on test sets. For success rate, we report both goal-finding (normal font) and request-fulfilling results (smaller grey font in parentheses). We also report the average number of different types of actions taken by the agent (across all task types).

| | Success Rate % ↑ | | | Avg. number of actions ↓ | | | |
|---|---|---|---|---|---|---|---|
| Agent | Unseen Start | Unseen Object | Unseen Environment | CUR | GOAL | SUB | DO |
| **No assistant and interaction policy** $\psi_\theta$ | | | | | | | |
| ($d^s$: current-state description, $d^g$: goal description) | | | | | | | |
| Sparse $d^s$ and $d^g$ | 43.4 (50.4) | 16.4 (23.2) | 3.0 (6.8) | - | - | - | 13.1 |
| Sparse $d^s$, dense $d^g$ | 67.2 (68.4) | 56.6 (58.2) | 9.7 (12.3) | - | - | - | 12.6 |
| Dense $d^s$, sparse $d^g$ | 77.9 (86.0) | 30.6 (40.3) | 4.1 (7.5) | - | - | - | 12.0 |
| Dense $d^s$ and $d^g$ | 97.8 (98.1) | 81.7 (83.3) | 9.4 (11.9) | - | - | - | 11.0 |
| **With assistant and interaction policy** $\psi_\theta$ | | | | | | | |
| Rule-based $\psi_\theta$ (baseline) | 78.8 (78.8) | 68.5 (68.5) | 12.7 (12.7) | 2.0 | 1.0 | 1.7 | 11.3 |
| RL-learned $\psi_\theta$ (ours) | 85.8 (86.8) | 78.2 (79.6) | 19.8 (22.5) | 2.1 | 1.0 | 1.7 | 11.1 |
| + Perfect nav. on sub-goals (skyline) | 94.3 (95.8) | 95.1 (96.1) | 92.6 (94.3) | 0.0 | 0.0 | 6.3 | 7.3 |

The human always returns dense descriptions. The interaction policy is trained in environments that are previously seen as well as unseen during pre-training. Finally, in the *evaluation* phase, the interaction policy is tested on three conditions: seen environment and target object type but starting from a new room (UNSEENSTR), seen environment but new target object type (UNSEENOBJ), and new environment (UNSEENENV). The pre-trained operation policy $\hat\pi$ is fixed during the training and evaluation phases. We create 82,104 examples for pre-training, 65,133 for training, and approximately 2,000 for each validation or test set. Details about the training procedure and the dataset are included in the Appendix.

## 6 RESULTS AND ANALYSES

**Settings.** In our main experiments, we set: the cost of taking a CUR, GOAL, SUB, or DO action to be 0.01 (we will consider other settings subsequently), the cost of calling DONE to terminate the main goal (i.e. task error) equal the (unweighted) length of the shortest-path from the agent's location to the goal, and the goal stack's size ($L$) to be 2. We compare our RL-learned interaction policy with a rule-based baseline that first takes the GOAL action and then randomly selects actions. In each episode, we enforce that the rule-based policy can take at most $\lfloor X_a \rfloor + y$ actions of type $a$, where $y \sim \text{Bernoulli}(X_a - \lfloor X_a \rfloor)$ and $X_a$ is a constant. We tune each $X_a$ on the validation sets so that the rule-based policy has the same average count of each action as the RL-learned policy. To prevent early termination, we enforce that the rule-based policy cannot take more DONE actions than SUB actions unless when its SUB action's budget is exhausted. We also construct a skyline where the interaction policy is also learned by RL but with an operation policy that executes subgoals perfectly. As discussed in §5, we are primarily interested in goal-finding success rate and will refer to this metric briefly as success rate.

**Main Results (Table 1).** To inspect the potential benefits of asking for additional information, we compute how much the operation policy $\hat\pi$ improves when it is supplied with dense information about the current and/or goal states. As seen, success rate of the operation policy is lifted dramatically when both the current-state and goal descriptions are dense ($\sim 2\times$ increase on UNSEENSTR, $\sim 5\times$ on UNSEENOBJ, and $\sim 3\times$ on UNSEENENV). We find that dense information about the current state is more helpful on UNSEENSTR, while dense information about the goal is more valuable on UNSEENOBJ and UNSEENENV. This is reasonable because on UNSEENSTR, the agent has been trained to find similar goals. In contrast, the initial goal descriptions in UNSEENOBJ and UNSEENENV are completely new to the agent, thus gathering more information about them is necessary.

Aided by our RL-learned interaction policy, the agent observes a substantial $\sim 2\times$ increase in success rate on UNSEENSTR, $\sim 5\times$ on UNSEENOBJ, and $\sim 7\times$ on UNSEENENV, compared to when performing tasks using only the operation policy. In unseen environments, with its capability of requesting subgoals, the agent impressively doubles the success rate of the operation policy that has access to dense descriptions. The RL-learned interaction policy is significantly more effective than

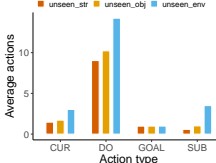
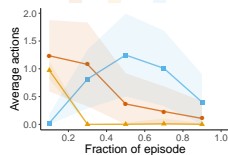

(a) How frequently does the interaction policy execute different actions, on average across different types of tasks? Subgoals are requested much more in unseen environments.

(b) Over the course of a trajectory, how does the frequency of different types of actions change in unseen environments? Subgoals are requested in the middle, goal information at the beginning.

Figure 2: Analyzing the behavior of the RL-learned interaction policy (on validation environments).

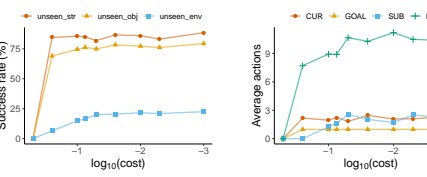

(a) Effect of cost on success rate.

(b) Effect of cost on action counts.

Figure 3: Analyzing the effect of simultaneously varying the cost of the CUR, GOAL, SUB, DO actions (on validation environments), thus trading off success rate versus number of actions taken.

the rule-based baseline (+7.1% on UNSEENENV). Compared to a policy that calls GOAL at the beginning and calls CUR at every step (which is equivalent to the dense-$d^s$-and-$d^g$ baseline), our policy achieves higher success rate on UNSEENENV while making two times fewer requests. This is due to the ability to request subgoals.

On UNSEENSTR and UNSEENOBJ, the RL-learned policy has not closed the gap with the operation policy that performs tasks with dense descriptions. Our investigation finds that limited information often causes the policy to not request information about the current state (i.e. taking CUR) and terminate prematurely or go to a wrong place. Encoding uncertainty in the current-state description (e.g., Finkel et al. (2006); Nguyen & O'Connor (2015)) is a plausible future direction for tackling this issue. Finally, results obtained by replacing the learned operation policy with one that behaves optimally on subgoals shows that further improving performance of the operation policy on short-distance goals would effectively enhance the agent's performance on long-distance goals.

**Behavior of the RL-Learned Interaction Policy.** Figure 2a characterizes behaviors of the RL-learned interaction policy in three evaluation conditions. We expect that tasks in UNSEENSTR are the easiest and those in UNSEENENV are the hardest. As the difficulty of the evaluation condition increases, the interaction policy issues more CUR, SUB, and DO actions. The average number of GOAL actions does not vary, showing that the interaction policy has correctly learned that making more than one goal-clarifying request is unnecessary. Figure 2b illustrates the distribution of each action along the length of an episode in the validation UNSEENENV dataset. The GOAL action, if taken, is always taken only once and immediately in the first step. The number of CUR actions gradually decreases over time. The agent makes most SUB requests in the middle of an episode, after its has attempted but failed to accomplish the main goals. We observe similar patterns on the other two validation sets.

**Effects of Varying Action Cost.** As mentioned, we assign the same cost to each CUR, GOAL, SUB, or DO action. Figure 3a demonstrates the effects of changing this cost on the success rate of the agent. Setting the cost equal to 0.5 makes it too costly to take any action, inducing a policy that always calls DONE in the first step and thus fails on all tasks. Overall, the success rate of the agent rises as we reduce the action cost. The increase in success rate is most visible in UNSEENENV and least visible in UNSEENSTR. Figure 3b provides more insights. As the action cost decreases, we observe a growth in the number of SUB and DO actions taken by the interaction policy. Meanwhile, the numbers of CUR and GOAL actions are mostly static. Since requesting subgoals is more helpful in unseen environments than in seen environments, the increase in the number of SUB actions leads the more visible boost in success rate on UNSEENENV tasks.

**Performing Tasks with Deeper Goal Stacks.** In Table 2, we test the functionality of our framework with a stack size 3, allowing the agent to request subgoals of subgoals. As expected, success rate on UNSEENENV is boosted significantly (+11.9% compared to using a stack of size 2). Success rate on UNSEENOBJ is largely unchanged; we find that the agent makes more SUB requests (averagely 4.5 requests per episode compared to 1.0 request made when the stack size is 2), but doing

Table 2: Success rates and numbers of actions taken with different stack sizes (on validation). Larger stack sizes significantly aid success rates in unseen environments, but not in seen environments.

| | Goal-finding success rate (%) ↑ | | | Average number of actions ↓ | | | |
| Stack size | Unseen Start | Unseen Object | Unseen Environment | CUR | GOAL | SUB | DO |
| --- | --- | --- | --- | --- | --- | --- | --- |
| 1 (no subgoals) | 92.2 | 78.4 | 12.5 | 5.1 | 1.9 | 0.0 | 10.7 |
| 2 | 86.9 | 77.6 | 21.6 | 2.1 | 1.0 | 1.7 | 11.2 |
| 3 | 83.2 | 78.6 | 33.5 | 1.3 | 1.0 | 5.0 | 8.2 |

so does not further enhance performance. The agent makes less CUR requests, possibly in order to offset the cost of making more SUB requests. Due to this behavior, success rate on UNSEENSTR declines with larger stack sizes, as information about the current state is more valuable for these tasks than subgoals. These results show that the critic model overestimates the $V$ values in states where SUB actions are taken, leading to the agent learning to request subgoals more than needed.

## 7 RELATED WORK AND CONCLUSION

**Transfer Learning in Reinforcement Learning.** Various frameworks have been proposed to model knowledge transfer from a more capable agent to a novice one (Da Silva & Costa, 2019). Torrey & Taylor (2013) introduce the action-advising framework where a learner strategically requests reference actions from a teacher. Da Silva et al. (2020) investigate uncertainty-based strategies for deciding when to request in this framework. In an agent-to-agent setting, (Da Silva et al., 2017; Zimmer et al., 2014; Omidshafiei et al., 2019) focus on learning a teaching policy in addition to an advice-requesting policy. An important assumption in these papers is that the teacher must share a common action space with the learner. More recent frameworks (Kim et al., 2019; Nguyen et al., 2019; Nguyen & Daumé III, 2019) relax this assumption by allowing the teacher to specify high-level subgoals instead of low-level actions. HARI can be viewed as a strict extension of these frameworks. It allows the human to specify not only subgoals, but also any additional information about the current state and the goal that agent can interpret. Moreover, HARI equips the agent with *multiple* communication intentions and teaches it to select the most useful intention to convey in a given situation. Another line of work employs standard RL communication protocol, where the human can only transfer knowledge through numerical scores or categorical feedback (Knox & Stone, 2009; Judah et al., 2010; Peng et al., 2016; Griffith et al., 2013). Maclin & Shavlik (1996) propose a framework where the human advises the agent using a domain-specific language, specifying rules that can be incorporated into the agent's model. In contrast, HARI operates with a black-box agent model. Sumers et al. (2020) extract features from various types of language feedback to construct a reward function for reinforcement learning. We instead focus on deployment-time communication and directly incorporate the human feedback as input to agent's operation policy.

**Task-Oriented Dialog and Generating Natural Language Questions.** HARI models a task-oriented dialog problem. Many variants of this problem requires the agent to compose specific questions (De Vries et al., 2017; Das et al., 2017; Thomason et al., 2020). The dominant approach in these problems is to mimic pre-collected human utterances. As discussed previously, naively mirroring human external behavior cannot enable agents to understand the limits of their knowledge. We teach the agent to understand its intrinsic needs through interaction with the human and the environment rather than through imitation of human behaviors. Another related line of work concerns generating natural language explanations of model decisions (Camburu et al., 2018; Hendricks et al., 2016; Rajani et al., 2019).

In summary, this paper presents a general POMDP framework for modeling human-agent communication. While we demonstrate this framework on a simplified navigation problem, our framework can theoretically capture richer types of human-agent communication. Hence, an important empirical question is how well our formulation generalizes to richer environments with more complex interactions and state spaces (Shridhar et al., 2020). Enhancing the sample efficiency of the learning policy by exploiting the hierarchical policy structure is an exciting future direction. Furthermore, techniques for generating faithful explanations (Kumar & Talukdar, 2020; Madsen et al., 2021) can be applied to enhance the specificity of the generated questions.

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

# A   TRAINING PROCEDURE

**Cost function.**   The cost function in our framework is given as follows

$$\bar{c}(s_t, G_t, \bar{a}_t) = \begin{cases} c(s_t, a_t^{\text{do}}) & \text{if } \bar{a}_t = \text{Do} \\ \gamma_{\bar{a}_t} & \text{if } \bar{a}_t \in \{\text{Cur}, \text{Goal}, \text{Sub}\}, \\ c(s_t, a_{\text{done}}) & \text{if } \bar{a}_t = \text{Done}, |G_t| = 1 \\ 0 & \text{if } \bar{a}_t = \text{Done}, |G_t| > 1, \end{cases} \tag{6}$$

**Subgoals.**   Let $p_t$ be the shortest path from the agent's current state $s_t$ to the current goal $g_t$, and $p_{t,i}$ be the $i$-th node on the path $(0 \le i < |p_t|)$. The subgoal location is chosen as $p_{t,k}$ where $k = \min(\lfloor |p|/2 \rfloor, l_{\max})$, where $l_{\max}$ is a pre-defined constant.

**Training Algorithms.**   We pre-train the operation policy $\hat{\pi}$ with DAgger (Ross et al., 2011), minimizing the cross entropy between its action distribution with that of a shortest-path oracle (which is a one-hot distribution with all probability concentrated on the optimal action).

We use advantage actor-critic (Mnih et al., 2016) to train the interaction policy $\psi_\theta$. This method simultaneously estimates an actor policy $\psi_\theta : \bar{\mathcal{B}} \to \Delta(\bar{\mathcal{A}})$ and a critic function $V_\phi : \bar{\mathcal{B}} \to \mathbb{R}$. Given an execution $\bar{\tau} = (\bar{s}_1, \bar{a}_1, \bar{c}_1 \cdots, \bar{s}_H)$, the gradients with respect to the actor and critic are

$$\nabla_\theta \mathcal{L}_{\text{actor}} = \sum_{t=1}^{H} \left( V_\phi(\bar{b}_t^v) - C_t \right) \nabla_\theta \log \psi_\theta(\bar{a}_t \mid \bar{b}_t^a) \tag{7}$$

$$\nabla_\phi \mathcal{L}_{\text{critic}} = \sum_{t=1}^{H} \left( V_\phi(\bar{b}_t^v) - C_t \right) \nabla_\phi V_\phi(\bar{b}_t^v) \tag{8}$$

where $C_t = \sum_{j=t}^{H} c_j$, $\bar{b}_t^a$ is a belief state that summarizes the partial execution $\bar{\tau}_{1:t}$ for the actor, and $\bar{b}_t^v$ is a belief state for the critic.

**Cost function.**   The cost function introduced in § 4.4 is not effective for learning the interaction policy because the task error is given only at the end of an episode. We extend the reward-shaping method proposed by Ng et al. (1999) to goal-conditioned policies, augmenting the original cost function with a shaping function $\Phi(s, g)$ with $s, g \in \mathcal{S}$. We set $\Phi(s, g)$ to be the (unweighted) shortest-path distance from $s$ to $g$. The cost received by the agent at time step $t$ is $\tilde{c}_t \triangleq \bar{c}_t + \Phi(s_{t+1}, g_{t+1}) - \Phi(s_t, g_t)$. We assume that the agent transitions to a special terminal state $s_{\text{term}} \in \mathcal{S}$ and remains there after it terminates execution of the main goal. We set $\Phi(s_{\text{term}}, \texttt{None}) = 0$, where $g_t = \texttt{None}$ signals that the episode has ended. Hence, the cumulative cost of an execution under the new cost function is

$$\sum_{t=1}^{H} \tilde{c}_t = \sum_{t=1}^{H} \bar{c}_t + \Phi(s_{t+1}, g_{t+1}) - \Phi(s_t, g_t) = \sum_{t=1}^{H} \bar{c}_t - \Phi(s_1, g_1) \tag{9}$$

Since $\Phi(s_1, g_1)$ does not depend on the action taken in $s_1$, minimizing the new cumulative cost does not change the optimal policy for the task $(s_1, g_1)$.

**Model Architecture.**   We adapt the V&L BERT architecture (Hong et al., 2020) for modeling the operation policy $\hat{\pi}$. Our model has two components: an encoder and a decoder; both are implemented as Transformer models (Vaswani et al., 2017). The encoder takes as input a description $d_t^s$ or $d_t^g$ and generates a sequence of hidden vectors. In every step, the decoder takes as input the previous hidden vector $b_{t-1}^s$, the sequence of vectors representing $d_t^s$, and the sequence of vectors representing $d_t^g$. It then performs self-attention on these vectors to compute the current hidden vector $b_t^s$ and a probability distribution over navigation actions $p_t$.

The interaction policy $\psi_\theta$ (the actor) is an LSTM-based recurrent neural network. The input of this model is the operation policy's model outputs, $b_t^s$ and $p_t$, and the embedding of the previously taken action $\bar{a}_{t-1}$. The critic model also has a similar architecture but outputs a real number (the $V$ value) rather than an action distribution. When training the interaction policy, we always fix the parameters of the operation policy. We find it necessary to pre-train the critic before training it jointly with the actor.

Table 3: Dataset statistics.

| Split | Number of examples |
|---|---|
| Pre-training | 82,104 |
| Pre-training validation | 3,000 |
| Training | 65,133 |
| Validation UNSEENSTR | 1,901 |
| Validation UNSEENOBJ | 1,912 |
| Validation UNSEENENV | 1,967 |
| Test UNSEENSTR | 1,653 |
| Test UNSEENOBJ | 1,913 |
| Test UNSEENENV | 1,777 |

**Representation of State Descriptions.** The representation of each object, room, or action is computed as follows. Let $f^{\text{name}}$, $f^{\text{horz}}$, $f^{\text{vert}}$, $f^{\text{dist}}$, and $f^{\text{type}}$ are the features of an object $f$, consisting of its name, horizontal angle, vertical angle, distance, and type (a type is either `Object`, `Room`, or `Action`; in this case, the type is `Object`). For simplicity, we discretize real-valued features, resulting in 12 horizontal angles (corresponding to $\pi/6 \cdot k, 0 \le k < 12$), 3 vertical angles (corresponding to $\pi/6 \cdot k, -1 \le k \le 1$), and 5 distance values (we round down a real-valued distance to the nearest integer). We then lookup the embedding of each feature from an embedding table and sum all the embeddings into a single vector that represents the corresponding object. For a room, $f^{\text{horz}}$, $f^{\text{vert}}$ $f^{\text{dist}}$ are zeroes. For an action, $f^{\text{name}}$ is either `ActionStop` for the stop action $a_{\text{done}}$ or `ActionGo` otherwise.

During pre-training, we randomly drop features in $d_t^s$ and $d_t^g$ so that the operation policy is familiar with making decisions under sparse information. Concretely, we refer to all features of an object, room or action as a *feature set*. For $d_t^s$, let $M$ be the number objects in a description. We uniformly randomly keep $m$ feature sets among the $M + 1$ feature sets of $d_t^s$ (the plus one is the room's feature set), where $m \sim \text{Uniform}(\min(5, M + 1), M + 1)$.

For $d_t^s$, we have two cases. If $g_1$ is not adjacent or equals to $s_1$, we uniformly randomly alternate between giving a dense and a sparse description. In this case, the sparse description contains the features of the target object and the goal room's name. Otherwise, with a probability of ⅓, we give either (a) a dense description (b) a (sparse) description that contains the target object's features and the goal room's name, or (c) a (sparse) description that describes the next ground-truth action.

We pre-train the operation policy on various path lengths (ranging from 1 to 10 graph nodes) so that it learns to accomplish both long-distance main goals and short-distance subgoals.

**Data.** Table 3 summarizes the data splits. From a total of 72 environments provided by the Matterport3D dataset, we use 36 environments for pre-training, 18 as unseen environments for training, 7 for validation UNSEENENV, and 11 for test UNSEENENV. We use a vocabulary of size 1738, which includes object and room names, and special tokens representing the distance and direction values. The length of a navigation path ranges from 5 to 10 graph nodes.

**Hyperparameters.** See Table 4.

Table 4: Hyperparameters.

| Hyperparameter Name | Value |
|---|---|
| **Environment** | |
| Max. subgoal distance ($l_{\max}$) | 3 nodes |
| Max. stack size ($L$) | 2 |
| Max. object distance for $d_t^s$ | 5 meters |
| Max. object distance for $d_t^g$ | 3 meters |
| Max. number of objects ($M_{\max}$) | 20 |
| Cost of taking each CUR, GOAL, SUB, DO action | 0.01 |
| **Operation policy $\hat{\pi}$** | |
| Hidden size | 256 |
| Number of hidden layers | 2 |
| Attention dropout probability | 0.1 |
| Hidden dropout probability | 0.1 |
| Number of attention heads | 8 |
| Optimizer | Adam |
| Learning rate | $10^{-4}$ |
| Batch size | 32 |
| Number of training iterations | $10^5$ |
| Max. number of time steps ($H$) | 15 |
| **Interaction policy $\psi_\theta$** | |
| Hidden size | 512 |
| Number of hidden layers | 1 |
| Entropy regularization weight | 0.001 |
| Optimizer | Adam |
| Learning rate | $10^{-5}$ |
| Batch size | 32 |
| Number of critic pre-training iterations | $5 \times 10^3$ |
| Number of training iterations | $5 \times 10^4$ |
| Max. number of time steps ($H$) | 30 |
| Max. number of time steps for executing a subgoal | $3\times$ shortest distance to the subgoal |

