# OpenReview forum: "Learning When and What to Ask: a Hierarchical Reinforcement Learning Framework"
_ICLR.cc/2022/Conference — ICLR 2022 Submitted_

### Official Review · Reviewer_LvSE · 2021-10-29

**Correctness:** 2
**Technical Novelty And Significance:** 2
**Empirical Novelty And Significance:** 1
**Recommendation:** 3
**Confidence:** 4

**Main Review:**

The authors present a "transfer learning" (TL) framework in which the agent is able to "ask for help" as one of the action options. While the authors propose an "unusual" modeling centered around POMDPs and robotic navigation (which could be good, bridging the gap between general TL for RL methods and this particular application), there has been a large amount of works with the same high-level idea in the last years. Yet, most of the related works are simply neglected and not even mentioned in the manuscript.

The survey below summarizes dozens of related works that follow this same general idea, and not even the Survey itself is cited:

Silva, Felipe Leno, and Anna Helena Reali Costa. "A survey on transfer learning for multiagent reinforcement learning systems." Journal of Artificial Intelligence Research 64 (2019): 645-703.

In particular, works from the "action advising" area are very relevant to the present manuscript. The seminal Teacher-Student framework [1] introduced the idea of proactively "asking for help" when needed in the RL context. The RCMP advising method [2], based on Adhoc Advising [3], propose a more modern takeaway on how to compute the agent "uncertainty" (very relevant to the manuscript at hand). Works that focus on the specific points claimed as contributions in the manuscript also exist, such as considering as a MDP the problem of learning "when to ask for help" [4, 5], or adding a hierarchical component to task selection before the low-level resolution of the task [6].

[1] Torrey, Lisa, and Matthew Taylor. "Teaching on a budget: Agents advising agents in reinforcement learning." Proceedings of the 2013 international conference on Autonomous agents and multi-agent systems. 2013.

[2] Silva, Felipe Leno, et al. "Uncertainty-aware action advising for deep reinforcement learning agents." Proceedings of the AAAI Conference on Artificial Intelligence. Vol. 34. No. 04. 2020.

[3] Silva, Felipe Leno, Ruben Glatt, and Anna Helena Reali Costa. "Simultaneously learning and advising in multiagent reinforcement learning." Proceedings of the 16th conference on autonomous agents and multiagent systems. 2017.

[4] Zimmer, Matthieu, Paolo Viappiani, and Paul Weng. "Teacher-student framework: a reinforcement learning approach." AAMAS Workshop Autonomous Robots and Multirobot Systems. 2014.

[5] Omidshafiei, Shayegan, et al. "Learning to teach in cooperative multiagent reinforcement learning." Proceedings of the AAAI Conference on Artificial Intelligence. Vol. 33. No. 01. 2019.

[6] Kim, Dong-Ki, et al. "Learning hierarchical teaching in cooperative multiagent reinforcement learning." (2019).

I would expect that several of those related works were included in the empirical evaluation. Yet, none of them are even mentioned in the paper! At the very least the distinction between the contribution of the present manuscript and those related works should be discussed in deep, as well as an explanation of why those methods were not added to the empirical evaluation.

Due to the aforementioned major omission, I cannot judge the contribution of the paper.



-------------------
After Rebuttal
-------------------

Although I agree the method has its merits and contribution, the manuscript does a poor job in situating the proposed work in the current state-of-the-art literature. Most of the discussion period was devoted to explaining where the method differs from current literature, which should have been clear since the beginning by reading the paper. I am also not entirely convinced the method is that different from the action advising framework. Sure, the present manuscript chooses to extract more information from the advice by enforcing the advisor to be a human (more restrictions = more information to be extract from the advice at the cost of less flexibility), while most of those methods are very general and focus on information that exist in any RL environment (i.e., states and actions). Yet, the same general idea holds, and the present method is not the first one that extracted from advice more than action suggestions. Those similarities should be embraced and throughfully explained in the manuscript, as in my view the proposed method is a specialized advisor-advisee framework for this specific situation that matters most to the authors. Therefore, my grades are maintained so that the authors can improve the manuscript for the next submission.

**Summary Of The Paper:**

The paper presents a reinforcement learning framework where the agent is able to "ask for help" to receive from a human additional information to more easily solve the task. The paper is kinda Robotic Navigation-oriented by the way tasks and "additional information" about the environment are described, as well as in the empirical evaluation. It's uncertain if the paper presents any significant contribution due to major omissions on the related works list (explanation in the main review)

**Summary Of The Review:**

Paper investigates an interesting and relevant topic. However, there is a major omission of related works. Therefore, my recommendation is to reject the paper.

---

> ### Author Response · Authors · 2021-11-17
> **Thank you the pointers and position of our work relative to prior work**
>
> We thank you very much for pointing us to the related work, which we unfortunately missed. We will include a discussion on this line of work in our revised version. Overall, these papers implement a traditional imitation learning communication protocol: the agent request action advice and the human gives actions that lie in the agent’s action space. Our framework implements a richer protocol, allowing the agent to convey more diverse types of intentions and the human to convey rich types of instructive and descriptive information (e.g., hints like “object X is next to object Y”, subtasks like “go to location Z”). Below is our detailed comparison with prior work.
>
> **Motivation: towards faithful and rich communication with humans at *deployment* time**
>
> Leveraging human assistance at deployment/test time is underexplored in ML, where the traditional focus is on independent task solving (“full autonomy”). Faithful and rich human-agent communication is helpful both for the agent and the human: leveraging human advice improves an agent’s task performance (empirically demonstrated in our paper), and being able to inform humans when and why it is about to fail makes an agent safer to use. Humans would not want agents that ignored them during their operation.
>
> Leveraging human assistance is not trivial, requiring effective communication with humans. Our paper tackles challenging communication problems: how can agents elicit good advice from humans? How do they know what to ask for? How can they incorporate rich types of human advice into their decisions?
>
> **Comparison with prior work: more faithful and richer communication**
>
> Two lines of work are closely related to our paper: one from RL/IL and the other from NLP. The RL/IL work includes papers you suggested (which we unfortunately missed, thank you for the pointers!) and [1] (cited). In the paper you suggested, the human-agent communication is **limited**: the agent can only ask for the next best action, and the human can only give actions that lie in the agent’s action space. More powerful frameworks [1] (cited by our paper) allows the human to give natural language advice, but the agent still requests generic help. On the other hand, in the NLP line of work [2], the agent can generate rich language utterances but it is not **not faithful**, in the sense that its language only mirrors pre-collected human utterances and does not necessarily represent what information is most useful to it.
>
> The human-agent communication in our framework is richer than in the RL/IL work and more faithful than in the NLP work. The human can transfer diverse types of information like hints (e.g., “object X is next to object Y”) and action advice that is not necessarily in the agent’s action space. The agent learns different types of intrinsic needs and faithfully conveys them to the human. We present a clean, rigorously defined POMDP-based formulation that can potentially facilitate theoretical studies in RL/IL and provide a firm skeleton for NLP researchers to enrich the communication with more natural language.
>
> **The generality of our framework**
>
> It is essential to distinguish our proposed framework (sections 2-4) from our experimental setting (section 5), which demonstrates only a practical scenario that the framework can model. Our framework allows passing rich information to the agent through state descriptions. As discussed in section 2 of the paper, “state description” is a general concept, encompassing any information about a state, including but not limited to visual perception and verbal description. The mechanism for the human to assist the agent is also general: the human gives information that connects with the agent's prior knowledge. This mechanism models how humans teach others *incrementally*. For example, to teach a learner to “make tea”, the teacher gives an instruction that refers back to the tasks that the learner is familiar with: “boil water, pour water into tea” (requesting new tasks corresponds to requesting new state descriptions in our framework).
>
> The bag-of-feature state descriptions in our experiments should not be viewed as the original sensor inputs that the agent perceives. Instead, they emulate only **the information extracted from the sensor inputs**. The sensor input can be arbitrarily rich. For example, in object-centric navigation, the agent is presented with an image of the current scene, from which it extracts object-related features for navigation. Working with these intermediate inputs allows us to easily control the amount of information fed to the agent, and focus on learning when and what to ask rather than having to deal with the nuisance in object recognition and language understanding.
>
> [1] Nguyen et al., 2019. https://arxiv.org/abs/1909.01871 and https://arxiv.org/abs/1812.04155
> [2] Camburu et al., 2018. https://arxiv.org/abs/1812.01193

---

### Official Review · Reviewer_9mjj · 2021-11-01

**Correctness:** 3
**Technical Novelty And Significance:** 2
**Empirical Novelty And Significance:** 1
**Recommendation:** 3
**Confidence:** 3

**Main Review:**

I am not convinced that this paper is ready for ICLR, it looks like this project is in an early stage with writing often unclear, results are presented without confidence intervals, in tables rather than figures. I am also not convinced of the strength of the contributions, even if the presentation was improved. The authors do a good job implementing existing algorithms in the context of the studied task, however the applicability of this framework seems to be very narrow. While reading this paper I kept wondering why is this a valuable research question, however I do not have a good answer.

A detailed list of my comments is below:

1. The writing is often unclear -- for example:

 - Page 1: "On every step, the agent can ask for more information about its current state, the goal state, or a subgoal state..." "The agent learns when to convey an intent to the assistant..."
this is confusing... so, does the agent convey intent, or ask questions? But the assistant is supposed to be omniscient, and always aware of the agent's intent?

 - Page 2:  states that agent has only a partial description of a goal state. So, does the uncertainty in this task reduce to uncertainty about the goal state? Toward the end of the paper I discovered that the uncertainty actually  comes from missing environment annotations. This needs to be clarified.

- Page 2: "The agent achieves this level of performance while issuing only 4.8 requests to the assistant on average, representing less than 1⁄4 of the total number of actions taken in a task execution." Why is this significant or relevant? The number of times the agent has to make requests is determined by how much information is missing in the evaluation task, compared to the task it was trained on -- which means that the exact numbers are arbitrary. Such references to arbitrary numbers happen in a few places through the paper.

2. Related work is very brief, and given in the end of the paper -- instead of at the beginning to motivate the current work. This makes it hard to evaluate the novelty of the contribution.
The goal stack maintenance section could be moved to the supplement to make space for motivation of this work.

3. It appears that there are multiple goals at the same time in the goal stack - if so, how does the agent arbitrate between multiple goals? I expected that there should be only one goal, which can have subgoals, but this is not clear in the text.

4. A computational experiment is used to show that the ability to request information was useful to the agent. However, it seems obvious that an agent with less uncertainty will do better. Section 6 describes the model comparison as comparing the agent that requests assistance to an agent that takes random actions. However, with missing annotations a rational agent that can not request information should still be able to rationally search the graph until the object is found - which seems a more reasonable  performance benchmark.  Alternatively, the authors could compare the current framework to a state-of-the-art agent that can solve the same benchmark problem. Was the indoor object-finding task used in previous work, or as part of benchmark problem sets?


5. Practical applications of this framework seem to be very limited, to a context of running RL agents with a near-perfect incomplete policy to evaluation tasks with missing annotations.

6. What is the point of using the Matterport3D dataset, as a basis for environment graphs? This 3D nature of this dataset, or that it is made to resemble a living space, is not in any way leveraged by the current algorithm.

7. The data presented in tables would be better readable as a plot.


**Summary Of The Paper:**

This paper formulates a hierarchical RL agent that learns to request information from human assistant, when task-critical information is missing. The assistant is assumed to be omniscent, present at all times, know the agent’s current state, know the agents goal state, and be able to accurately provide the requested information. On each step the agent has 5 types of actions: it can either issue one of the three information queries (request current state description, goal state description, subgoal description), or execute the highest value action according to its beliefs, or terminate execution. Such a framework can be applicable to situations where an RL agent is trained on a certain task, but deployed on a similar task with some information missing (for example, a missing object annotation).
The framework is demonstrated on simulated human-assisted object finding task.  The paper shows that an agent equipped with a human assistant achieves an improved rate of successfully completing the task, compared to an unassisted agent.



**Summary Of The Review:**

I am not convinced that this paper is ready for ICLR, it looks like the project is in an early stage. The writing is unclear, results are presented in tables rather than figures -- and are mostly descriptions of model performance rather than model comparisons. I am also not convinced of the strength of the contributions, even if the presentation is improved. The authors do a good job implementing existing algorithms in the context of the given task, however the applicability of this framework is very narrow, to a context of running RL agents with a near-perfect incomplete policy to evaluation tasks with missing annotations. The paper claims that the agent that can request assistance from a human to reduce uncertainty performs better on the given task, than an agent that can not request assistance -- this claim is correct, however it also seems obvious.

---

> ### Author Response · Authors · 2021-11-17
> **Applicability of our framework and answers to specific questions**
>
> We thank you for your comments. We strongly encourage you to read our general response in addition to this response.
>
> **Applicability of the framework**: it is essential to distinguish our proposed framework (sections 2-4) from our experimental setting (section 5), which demonstrates only a practical scenario that the framework can model. Our framework specifies a general communication protocol between the agent and the human. The information sent from the human to the agent is encoded as a “state description”, which can contain any type of information about the state (footnote 1).  This protocol is much richer than those in previous work. For example, in work suggested by reviewer LvSE, the agent can only ask for low-level action advice from the human (the traditional imitation learning protocol).
>
> We require the agent to have some built-in knowledge (the pre-trained policy). This knowledge represents a common ground with humans. **Without a common ground, communication cannot happen**. You cannot advise an agent if it does not understand your advice.
>
> The term “missing annotations” gives a connotation of narrowness and does not accurately reflect the nature of the information sent by the human. The human assists the agent by giving information that **connects to its prior knowledge**. This captures typical scenarios that could be found in a wide range of applications. For example, when a robot does not know how to “make coffee”, the human gives an instruction “boil water, pour it into a cup...”. The instruction is helpful when it refers to the tasks the robot already knows how to perform. You could view the instruction as “missing annotations”: by saying “make coffee”, the human *implicitly* means “make coffee [by boiling water…]”. The robot then may not realize the cup mentioned in the instruction. The human may say “the cup is on the left” but, again, this is only helpful if the agent knows what to do if there is a cup nearby.
>
> The bag-of-feature representation in our experimental setting may have given the impression that our framework works only with missing features. These features should not be viewed as the raw input fed to the agent; it represents the **information the agent extracts from the raw input**. For example, in object-centric navigation, the agent is presented with an image of the current view, from which it extracts object features for navigation; similarly, we imagine that the agent extracts only relevant features from human utterances for its decision. We use this discrete representation to (1) easily simulate different amounts of input information and (2) focus on learning when and what to ask rather than object recognition and language understanding.
>
> Answers to specific questions:
>
> 1. As stated in the intro, “[the] agent is equipped with .... intents that can be easily translated into human-intelligible requests”. Asking a question is to convey an information-seeking intention.
>
> We never assume the human to be omniscient: they know the agent’s *environment* state and goal, but they do not know its inner state.
>
> The agent also has imperfect information about its *current state*, not just the goal state (sec 5).
>
> "¼ actions are requests": we are comparing our agent with the agent that asks all the time (the dense ds and dg skyline). Our agent outperforms this agent on unseen environments, while making fewer requests. We will revise this claim.
>
> 2. We can add an explicit related work section to the revised version. Please see the “prior work” section of our general response.
>
> 3. As stated in figure 1, the agent always executes the task at the top of the stack. We will make this more clear.
>
> 4. We are not sure what “rationally search the graph” means. Our "no assistance" baselines search the graph with pre-learned knowledge. An approach that exhaustively searches the graph overly exploits the structure of the simulator and may not scale to real continuous environments. We impose a time constraint so the agent cannot simply visit an entire house for each task. Even if the agent performed an exhaustive search, it would still need to detect objects (i.e. learning when to call DONE).
>
> This navigation problem is proposed by our paper. We implement a transformer-based model trained with imitation learning, which is a strong baseline for the popular VLN problem (https://arxiv.org/abs/2011.13922). Our task is harder than VLN in that the task command is high-level (“find O in room R” vs “turn left... go to X... stop there”).
>
> 5. Discussed at the beginning of this response.
>
> 6. The positions of the objects and their co-occurrences are encoded in the information given to the agent (i.e. the fact that an oven is in a kitchen). What is missing is just the visual appearance of the scenes.
>
> 7. Could you elaborate on this point? Which table(s) do you suggest us convert to a plot and what about a plot would make it clearer? We are happy to adjust them if that makes the paper better.

---

> > ### Author Response · Authors · 2021-11-22
> > **Response to "the gain of the agent that can request human assistance seems obvious"**
> >
> > Replying to your comment that the gain from leveraging human assistance seems obvious, we'd also like to add that while we should expect that human assistance would benefit the agent, **extracting** this benefit is not trivial. That requires the agent to be able to communicate effectively with the human, being to elicit good advice from the human, to interpret the advice, and to incorporate the interpreted information into its decision-making process. Our contribution is to construct a practical model of these processes. With our model of communication, the agent cannot interact with the human to leverage their assistance. Furthermore, the idea of reducing uncertainty of the agent on new tasks is general and modeled after real-world scenarios (as discussed in our previous response to you). In order perform well on a new task, the agent can only either (i) self-improve its policy or (ii) rely on the human to demonstrate that the current task can be transformed or decomposed into tasks that the agent is more familiar with.

---

### Official Review · Reviewer_rgjt · 2021-11-03

**Correctness:** 3
**Technical Novelty And Significance:** 3
**Empirical Novelty And Significance:** 3
**Recommendation:** 5
**Confidence:** 4

**Main Review:**

**Pros**:
* Writing and Presentation: The paper is well structured. All the graphs were well annotated.
* Modelling: The POMDP designed to obtain the interaction policy was succinct and rich enough to represent the assistive task. It presents a framework that can be extended further/utilized in several other multi-agent problems and opens up avenues for future research.
* Goal Stack and using multiple Sub-goals: The idea of introducing a goal stack and including sub-goals as a possible action for the interaction policy was very interesting. The generalization capabilities of the method on unseen environments also look promising.
* Limitations of the work are well addressed.

**Cons**:
* Experimental Evaluation: The experiments done on the Matterport3D simulator under three different conditions was exhaustive. However, it would have helped to know a few more details on the SUB action: how was the subgoal given to the agent during training -- particularly when the subgoal location was not adjacent /coincides with the agent's current location.  I did not fully understand this action as presented by the human during training. Experimental details were missing and given that the problem is heavily dependent on modeling the POMDP, it would have helped to know the details of the setup elaborately.

* Baselines: The rule-based baseline seemed to be particularly weak in the context of this work. Perhaps a slightly stronger baseline -- including some more domain knowledge could have led to interesting ablation studies.

* Typos in Notation: Typo on Page 4, where the subgoal function was first introduced -- the symbol used was \sigma_{A}.


**Additional Comments**:

I find the premise of the work very interesting and was curious to know if there were any thoughts on how the presented algorithm/model would perform (robustness of the technique to noise) in case there was noise induced in the feedback from the assistant -- like the assistant presented a noisy estimate of the current state description or a possible sub-goal?


**General Comments**:
Overall I liked the approach of learning interaction policies using the hierarchical RL and think that it presents a good starting point for future research in the direction of learning policies under assistance. The success rate on unseen tasks guided by sub-goals is particularly promising. There could have been some more clarity in the experimental setup (for training), and additional baselines used for evaluation.

*Originality*:  Moderate

*Clarity*: Moderate

*Quality*: Good

*Significance*: Moderate to High

**Summary Of The Paper:**

The paper operates in a 2-agent setting, where one agent is a learning agent and the other agent is the assistant. The objective is for the learning agent to seek help from the assistant while solving a task, and the decision of what to ask the assistant and when to do so is made by learning an interaction policy using hierarchical reinforcement learning -- in particular using a variant of POMDPs. The assumption is that the learning agent's operational policy on the task environment is not changed and an additional interaction (hierarchical) policy is learned on top of the operational policy which dictates when and what to ask the assistant. The authors have empirically evaluated their approach on a (simulated) human-assisted navigation task, and show that the proposed method achieves close to seven times an increase in the success rate of solving the task compared to the policy obtained by learning without assistance. Additionally, the results also show that the frequency of seeking assistance was only 25 percent of all actions executed to solve the task.

**Summary Of The Review:**

The paper presents a good starting point for future research in the area and a model for application to several real-world problems such as assisted teaching (when to provide students help and how, for instance). However, the experimental section could be made clearer since the results presented look promising. An additional stronger baseline (though I understand this might be non-trivial to formulate) would have made the claims stand out. At the moment, both the baselines are weak (no assistance/rule-based assistance). Hence. I am inclined to reject the paper at the moment.

---

> ### Author Response · Authors · 2021-11-17
> **Thank you and more details about the experimental setting**
>
> We thank you for your appreciation of the papers. We strongly encourage you to read our general response in addition to this specific response.
>
> **Detail about the SUB action**: in the original submission, we provided the detail about how the SUB action is given as footnote 4 (page 6). We will make this information more prominent in the revised version. Let p be the path from the current location to the goal and |p| is its length. The subgoal is the k-th node on this path, where k = min(|p| / 2, d), where d is a predefined constant (d = 3 in our experiments). The description of the subgoal is the “dense description” of the k-th node (“dense description” is defined in section 5).
>
> **Baselines with domain knowledge**: we are not sure whether by “domain knowledge”, you referred to common sense about asking questions or common sense about the environment. If you have a concrete baseline that you’d like us to implement, please let us know. Our “rule-based” baseline in fact encodes substantial knowledge about the optimal strategy: it always asks for goal clarification first. To prevent early termination, we enforce that the rule-based policy *cannot take more DONE actions than SUB actions unless when its SUB action’s budget is exhausted*. In other words, this baseline does not terminate prematurely and uses up all its budget of subgoal requests. Hence, our learned policy can only outperform this baseline by smartly choosing *when* to take the CUR and SUB actions and *which* one to take. This is exactly the capability we want to evaluate. We will include more details about the rule-based baseline in the revised version.
>
> **Noisy teacher**: we thank you for this suggestion. We could inject random noise to the teacher to evaluate robustness, though we imagine the performance would certainly go down. An alternative direction is to model the noise after real human behavior, which is beyond the scope of this paper and we will leave it for future work.

---

> > ### Comment · Reviewer_rgjt · 2021-11-20
> > **Clarifications regarding the Experimental Framework**
> >
> > Thank you for clarifying the dynamics of the SUB action, and the rule-based baseline. By `domain knowledge' I was referring to common sense about the environment. But I see the working of the baseline now, after the explanation.

---

> > > ### Comment · Reviewer_rgjt · 2021-12-01
> > > **Thanks for the revisions**
> > >
> > > Thanks for revisions! After following all the discussions, I will keep my score.

---

### Author Response · Authors · 2021-11-17
**General response: motivation, comparison with prior work, and generality**

**Summary**

We thank the reviewers for your comments on the paper. Overall, we received mixed opinions. **Reviewer rgit** states that *“the paper is well structured. All the graphs were well annotated”*, and that our formulation is *“succinct and rich enough to represent the assistive task”* and *“opens up avenues for future research”*. **Reviewer 9mjj** holds a more critical opinion, finding that *“[the] applicability of this framework seems to be very narrow”* and the presentation of the paper is *“unclear”*. **Reviewer LvSE** *“cannot judge the contribution of the paper”*.

We hope that our response can help the reviewers reach a consensus more easily. Our response focuses on (i) positioning our contributions with prior work (ii) clarifying the generality of our general framework beyond the experimental setting.

**Motivation: towards faithful and rich communication with humans at *deployment* time**

Leveraging human assistance at deployment/test time is underexplored in ML, where the traditional focus is on independent task solving (“full autonomy”). Faithful and rich human-agent communication is helpful both for the agent and the human: leveraging human advice improves an agent’s task performance (empirically demonstrated in our paper), and being able to inform humans when and why it is about to fail makes an agent safer to use.

Leveraging human assistance is not trivial, requiring effective communication with humans. Our paper tackles challenging communication problems: How can agents elicit good advice from humans? How do they know what to ask for? How do they incorporate rich types of human advice?

**Comparison with prior work: more faithful and richer communication**

Two lines of work are related to our paper: one from RL/IL and the other from NLP. The RL/IL work includes papers suggested by reviewer LvSE (which we unfortunately missed, thank you for the pointers!) and [1] (cited). In the papers suggested by reviewer LvSE, the human-agent communication is **not rich**: the agent can only ask for the next best action, and the human can only give actions in the agent’s action space. More powerful frameworks [1] (cited by our paper) allows the human to give natural language advice, but the agent still requests generic help. On the other hand, in the NLP line of work [2], the agent can generate rich language utterances but it is not **not faithful**, in the sense that its language only mirrors pre-collected human utterances and does not necessarily specify what information is most useful to it.

The human-agent communication in our framework is richer than in the RL/IL work and more faithful than in the NLP work. The human can transfer diverse types of information like hints (e.g., “object X is next to object Y”) and action advice that is not necessarily in the agent’s action space. The agent learns different types of intrinsic needs and faithfully conveys them to the human. We present a clean, rigorously defined POMDP-based formulation that can potentially facilitate theoretical studies in RL/IL and provide a firm skeleton for NLP researchers to enrich the communication with more natural language.

**The generality of our framework**

It is essential to distinguish our proposed framework (sections 2-4) from our experimental setting (section 5), which demonstrates only a practical scenario that the framework can model. Our framework allows passing rich information to the agent through state descriptions. As discussed in section 2 of the paper, “state description” is a general concept, encompassing any information about a state, including but not limited to visual perception and verbal description. The mechanism for the human to assist the agent is also general: the human gives information that connects with the agent's prior knowledge. This mechanism models how humans teach others *incrementally*. For example, to teach a learner to “make tea”, the teacher gives an instruction that refers back to the tasks that the learner is familiar with: “boil water, pour water into tea” (requesting new tasks corresponds to requesting new state descriptions in our framework).

The bag-of-feature state descriptions in our experiments should not be viewed as the original sensor inputs that the agent perceives. Instead, they emulate only **the information extracted from the sensor inputs**. The sensor input can be arbitrarily rich. For example, in object-centric navigation, the agent is presented with an image of the current scene, from which it extracts object-related features for navigation. Working with these intermediate inputs allows us to easily control the amount of information fed to the agent, and focus on learning when and what to ask rather than having to deal with the nuisance in object recognition and language understanding.

[1] Nguyen et al., 2019. https://arxiv.org/abs/1909.01871 and https://arxiv.org/abs/1812.04155
[2] Camburu et al., 2018. https://arxiv.org/abs/1812.01193

---

### Author Response · Authors · 2021-11-22
**Changes in the rebuttal revision**

We have incoporated the feedback from the reviewers in the rebuttal revision:

High-level changes:

- Update the introduction to help the reviewers better understand our motivation and the significance of our work.

- Extend the related work section to better position our work in the literature (especially incorporating the work suggested by reviewer LvSE)

- Join the appendix with the main paper (previously it was submitted in a separate file).

- Provide new examples in section 3 to better illustrate the generality of the idea of the human "giving advice that connects to the agent's prior knowledge" (addressing the "narrow" concern of 9mjj).

Low-level changes:

- Reviewer rgjt: we move the detail about the subgoal action to the appendix, and add more details about the rule-based baseline in "Settings" (section 6).

-  Reviewer 9mjj: in section 6 (results), we elaborate our claim about the agent's asking efficiency (the "1/4 of actions are requests" claim). We replace that claim with a more general claim in the intro and abstract. We explain how the agent executes goals in its stack in section 4.2.

---

### Decision · Program_Chairs · 2022-01-20

**Decision:**

Reject

**Comment:**

This paper is on the theme of active reinforcement learning with a human/assistant in the loop. Under partial observability, an agent acts as per an interaction policy that gathers state/goal information from the assistant, while an operational policy assumed pre-learnt in this paper executes low-level actions.  The reviews acknowledge the relevance of this topic and that the paper is well structured and coherently presented overall. However, there are unanimous concerns around experimental evaluation being unconvincing, lack of strong baselines and lack of thorough coverage of related work precluding an accurate assessment of claimed contributions. As such, the paper is not in a form that can be accepted at ICLR --  the authors are encouraged to revise their submission as per review feedback.